# Effect of Batch Dissimilarity on Permeability of Stacked Ceramic Foam Filters and Incompressible Fluid Flow: Experimental and Numerical Investigation

Shahin Akbarnejad *, Anders Tilliander, Dong-Yuan Sheng and Pär Göran Jönsson

Department of Materials Science and Engineering, Royal Institute of Technology (KTH),
S-100 44 Stockholm, Sweden; anderst@kth.se (A.T.); shengdy@kth.se (D.-Y.S.); parj@kth.se (P.G.J.)
* Correspondence: shahinak@kth.se

**Abstract:** Ceramic foam filters (CFFs) are used to remove inclusions and/or solid particles from molten metal. In general, the molten metal poured on the top of a CFF should reach a certain height to form the pressure (metal head) required to prime the filter. For estimating the required metal head and obtaining the permeability coefficients of the CFFs, permeability experiments are essential. Recently, electromagnetic priming and filtration of molten aluminum with low and high grades of CFF, i.e., 30, 50 and 80 pore per inch (PPI) CFFs, have been introduced. Since then, there has been interest in exploring the possibility of obtaining further inclusion entrapment and aluminum refinement by using electromagnetic force to prime and filter with stacked CFFs. The successful execution of such trials requires a profound understanding concerning the permeability parameters of the stacked filters. Such data were deemed not to exist prior to this study. As a result, this study presents experimental findings of permeability measurements for stacks of three 30, three 50 and three 80 PPI commercial alumina CFFs from different industrial batches and compares the findings to numerically modelled data as well as previous research works. Both experimental and numerical findings showed a good agreement with previous results. The deviation between the experimentally and numerically obtained data lies in the range of 0.4 to 6.3%.

**Keywords:** stacked ceramic foam filters; alumina CFF; porous media; filtration; permeability

## 1. Introduction

Ceramic foam filters (CFFs) are made of open cell material [1–7] that, depending on the type, can possess relatively high mechanical properties such as high strength and structural uniformity, high thermal and chemical resistance and resistance to creep, etc. [1,3,5–11]. In addition to distinct mechanical properties, CFFs have a porous and tortuous structure and low resistance to fluid flow [4–6,8–10,12]. Such remarkable properties make them very attractive in different industries, as well as for various applications in the metallurgy, chemical and automotive industries [1–4,7,9–16]. In metallurgy, and particularly in the aluminum industry, CFFs are used to remove inclusions and/or solid particles from molten metal [3,4,13,17–25]. In general, it is necessary for the molten metal poured on the top of a CFF to reach a certain height to form the pressure (metal head) or gravitational force required to prime the filter [2,26]. Here, priming is defined as filling the filter with molten metal and removing the entrapped air [27–29]. For estimating the required metal head and for estimating, adjusting and/or maintaining the pressure gradient required to achieve the desired flow rate, it is necessary to carry out permeability experiments to obtain the permeability coefficients of the filter [2,11,19,26,29–35].

Fluid flow through porous media is usually defined by Darcy's empirically obtained equation, also known as Darcy's law, i.e., Equation (1) [13,31,33,34,36–41]. In this equation, pressure gradient $\Delta P$ (Pa) across the porous medium $L$ (m) is directly proportional to the fluid velocity $V$ (m/s) and inversely proportional to a coefficient $k$, i.e., the Darcy

permeability coefficient [13,31,33,34,36–40]. This equation, however, is only applicable at very low flow rates and small Reynolds numbers, i.e., $R_e \leq 1$ [13,33,34,36,37,39,41]. Therefore, by increasing the fluid velocity and increasing the Reynolds number, the pressure drop becomes nonlinear, and Darcy's law cannot describe the flow [13,31,33,34,38,39,41–43]. In such cases, for an incompressible fluid flow through a homogenous porous medium, the Forchheimer equation, i.e., Equation (2), can be used [2,6,13,19,26,29,31,36,39,41].

$$\frac{\Delta P}{L} = \frac{V}{k} \tag{1}$$

$$\frac{\Delta P}{L} = \frac{\mu V_s}{k_1} + \frac{\rho V_s^2}{k_2} \tag{2}$$

where $\mu$ (Pa·s) is the fluid dynamic viscosity, $V_s$ (m/s) is the superficial velocity, $\rho$ (kg/m$^3$) is the fluid density and $k_1$ (m$^2$) and $k_2$ (m) are the Darcy and non-Darcy permeability coefficients.

Recently, Kennedy [19,22,27,30,44] and Fritzsch [21,28,45,46] demonstrated the possibility of electromagnetic filtration of molten aluminum by using ceramic foam filters as well as priming stacks of CFFs. Since then, there has been interest in exploring the possibility of obtaining further inclusion entrapment and refinement in molten aluminum with electromagnetic forces. The successful execution of such an investigation requires an experimentally backed understanding of the permeability parameters, i.e., the Darcy and non-Darcy coefficients, in the stacked filters. Such data were deemed not to exist prior to this study.

This work aimed to obtain pressure gradients, as well as Darcy and non-Darcy permeability coefficients, for three stacked 30, three stacked 50 and three stacked 80 PPI alumina CFFs from different industrial batches and for several fluid flow rates. Furthermore, we aimed to compare the obtained data to previous research data. In addition, the liquid permeability experiments were numerically modelled to validate and explain the experimental findings.

## 2. Materials and Method

### 2.1. Liquid Permeability Experiments

Liquid permeability characteristics of stacks of three ~50 mm thick commercial alumina ceramic foam filters (CFFs) for 30, 50 and 80 PPI were experimentally obtained with water as the working fluid. A Plexiglas filter holder was constructed and used to hold the filters during the experiments, as shown in Figure 1.

Nine samples, three ~51 mm diameter samples for each PPI grade, were cut and prepared from ~50 mm thick, standard, 9 inch, square ceramic foam filters using a computer numerical control (CNC) water jet machine. After cutting, the samples were manually resized to ~49.5 mm diameter to fit into the filter holder [3]. Then, the dimensions were obtained by using a digital caliper with an accuracy of 0.03 mm and a resolution of 0.01 mm. The total and open pore porosities of the filters were calculated based on weight and volume [3,19], i.e., Equations (3) and (4).

$$\text{Total Porosity} = \frac{(\text{Mt} - \text{Ma})}{\text{Mt}} \times 100 \tag{3}$$

$$\text{Open Pore Porosity} = \frac{(\text{Vt} - \text{Va})}{\text{Vt}} \times 100 \tag{4}$$

where Mt (g) is the theoretical weight, Ma (g) is the measured weight, Vt (mm$^3$) is the theoretical volume and Va (mm$^3$) is the measured volume.

The theoretical volumes of the samples were calculated based on the measured dimensions of the samples. Then, the theoretical weight was estimated by considering the density of the CFF, which was obtained from the manufacturer, 3.48 g/cm$^3$. Later, the samples were weighed using a digital laboratory scale with a resolution of 0.01 g to obtain the weight. In addition, the actual volume was measured using a series of 50, 100 and 500 mL

graduated cylinders with accuracies in the range of 0.075 to 0.75 mL, as explained in detail elsewhere [29] and presented in Table 1. The difference in the total and open pore porosity values is expected to be less than 5 percent in alumina CFFs [3,8,19].

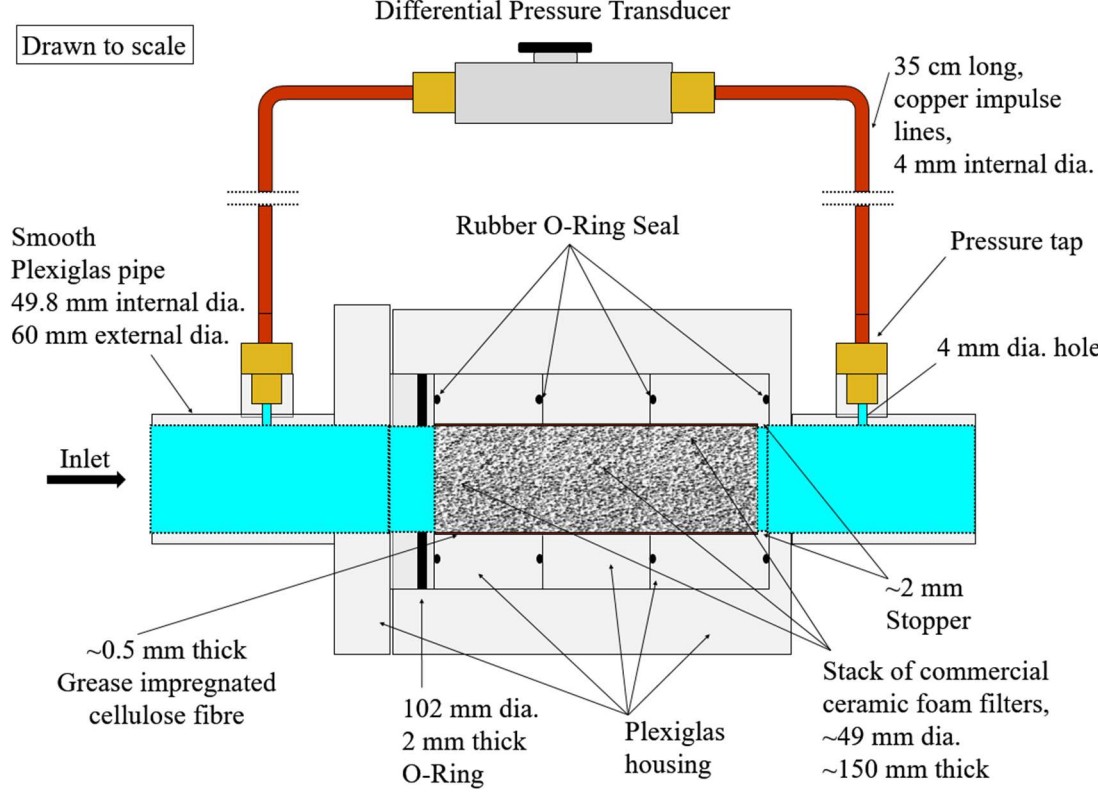

**Figure 1.** Experimental apparatus.

**Table 1.** Filter dimension and porosity values data from [29].

| Filter | | Diameter | Thickness | Total Porosity | Open Pore Porosity |
| No. | Type | (mm) | (mm) | (%) | (%) |
| --- | --- | --- | --- | --- | --- |
| N1 | 30 PPI | 49.33 ± 0.30 | 50.42 ± 0.07 | 90.1 | 88.8 |
| N2 | 30 PPI | 49.00 ± 0.37 | 50.83 ± 0.04 | 90.8 | 90 |
| N3 | 30 PPI | 49.38 ± 0.14 | 50.76 ± 0.06 | 90.1 | 91.5 |
| N1 | 50 PPI | 49.58 ± 0.18 | 50.88 ± 0.05 | 85.8 | 83.5 |
| N2 | 50 PPI | 49.30 ± 0.17 | 49.98 ± 0.02 | 86.1 | 84.6 |
| N3 | 50 PPI | 49.68 ± 0.10 | 50.63 ± 0.06 | 85.9 | 82.6 |
| N1 | 80 PPI | 49.63 ± 0.15 | 49.79 ± 0.04 | 85.6 | 81.5 |
| N2 | 80 PPI | 49.38 ± 0.28 | 50.28 ± 0.03 | 86.4 | 85.8 |
| N3 | 80 PPI | 49.30 ± 0.15 | 50.96 ± 0.06 | 87.1 | 85.1 |

To avoid fluid bypassing during permeability experiments, samples were carefully sealed. It has been shown recently that not sealing can cause up to 60% deviation in the experimentally obtained pressure gradients compared to those of fully sealed single filters [2]. Meanwhile, the importance of proper sample sealing has also been emphasized in the literature [5,8,10,19,20,32,33,35,47–53]. The sealing procedure includes three main steps: (i) Blinding, i.e., blocking of the sidewalls of the samples, (ii) resizing and (iii) wrapping in grease-impregnated cellulose fibers to tighten the samples once wet, after fitting them into the filter holder. The details of sample preparation and sealing can be found elsewhere [29]. Figure 2 illustrates samples made from a 50 PPI CFF, as well as the fully sealed samples fitted into the filter holders. During the experiments, filter N1 was always kept at the inlet, filter N2 in the middle and filter N3 was placed towards the outlet of the apparatus. In addition, the CFFs were touching each other, and no gap was introduced between them.

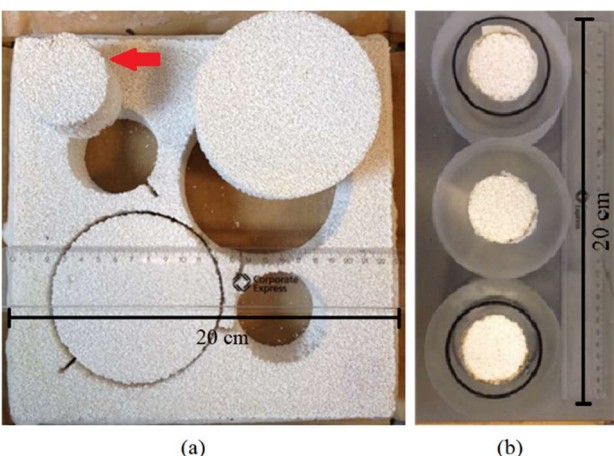

**Figure 2.** (**a**) a 9 inch 50 PPI filter with a sample taken from the filter and (**b**) the fully sealed samples fitted into the holders.

Ordinary tap water in the temperature range of 282 K to 284 K (9 °C to 11 °C) was circulated through a 49.8 mm smooth pipe using a submersible pump producing velocities from 0.03 to 0.4 m/s [26,29]. The pump was placed at the bottom of a 53 kg capacity container. A DN 25 ball valve was used to regulate the flow. The valve was located between the outlet of the pump and the inlet of the experimental setup pipeline. A FLUKE 80PT-25 T-Type probe with an accuracy of $\pm 1$ °C in 0 °C to 350 °C, together with a NI USB-TC01 data logger, was also placed in the container to measure and log the water temperature during the experiments. A minimum of 13 L/D is required to obtain a fully developed turbulent flow profile [39]. Here, a straight 1.2 m inlet pipe, or about 25 L/D, was used before the experimental apparatus, and a similar length of straight pipe was used after the apparatus. A pressure transducer, DF-2 (AEP, Transducer, Italy), was used to measure the differential pressure. This equipment had a pressure measuring range from 0 to 1 bar, a 4 to 20 mA DC output range and a certified error of ‹$\pm$0.04% of reading based on the factory calibration. The current produced by the pressure transducer was measured by a handheld FLUKE 289/FVF true-RMS digital multi-meter with a resolution of 0.001 mA in a 50 mA DC range and a data-logging feature.

The water velocity was calculated based on the mass flow measured during a specified time using the weight gain in a second container with a maximum capacity of 53 kg of water. To be specific, the container was placed at the end of the experimental setup and on an OHUAS T31P scale (3000 series indicator) equipped with the OHAUS Data Acquisition Software (DAS). Overall, the data were collected at one-second intervals for all three measurements and logging devices [29]. Later, the collected data were used to calculate the water velocity, density and dynamic viscosity.

The Thiesen–Scheel–Diesselhorst [54] equation (Equation (5)) was used to calculate water density as a function of temperature, where $T$ is temperature in °C and $\rho$ (kg/m$^3$) is the water density. To calculate the water velocity, first the mass flow rate (kg/s) was obtained, i.e., by plotting the weight gain at one-second intervals and calculating the slope of the linear regression equation. Then, the obtained mass flow rate was divided by the calculated average water density and inner cross section area of the pipe to determine the water velocity (m/s). Later, the water viscosity was calculated as a function of temperature and density according to the recommended formulation by the International Association for the Properties of Water and Steam (IAPWS) for the viscosity of ordinary water [55,56].

$$\rho = 1000 \left[ 1 - \frac{T + 288.9414}{508,929.2 \times (T + 68.12963)} (T - 3.9863)^2 \right] \tag{5}$$

The experimentally obtained pressure drop data, fluid flow rates and the calculated values of the fluid viscosities and densities were used to obtain the Darcy $k_1$ and non-

Darcy $k_2$ permeability coefficients based on Forchheimer's equation, i.e., Equation (2). Several methods were evaluated to calculate the permeability coefficients, as explained elsewhere [26,29]. Ergun's approach, i.e., dividing the Forchheimer equation (Equation (2)) by velocity and performing linear regression, provided the lowest average deviation of the experimentally obtained pressure gradients [2,19,26,29]. Therefore, Ergun's approach was selected as the most appropriate method for obtaining the Darcy $k_1$ and non-Darcy $k_2$ coefficients. The empirically obtained Darcy $k_1$ and non-Darcy $k_2$ permeability coefficients of the recent work were then compared to the findings of the previous works by Akbarnejad et al. [26,29], Kennedy et al. [19] and Zhang [57].

## 2.2. Numerical Modelling

To obtain the pressure gradient as a function of fluid velocity using computational fluid dynamics (CFD) and to compare the CFD results to the experiment data, 3-dimensional axisymmetric stacks of three 30, three 50 and three 80 PPI experimental setup models were created. The 3D models simulated perfectly sealed filters where no gap exists between the filter media and filter holders. An inlet pipe was connected to the top of the first filter and an outlet pipe to the lower part of the third filter. As a result, the simulated fluid, i.e., water, entered from the upper surface of the first filter, flowed through all three filters and exited from the lower side of the third filter to the outlet, as shown in Figure 3. Consequently, the fluid did not leave or enter the stacked filters from the sidewalls.

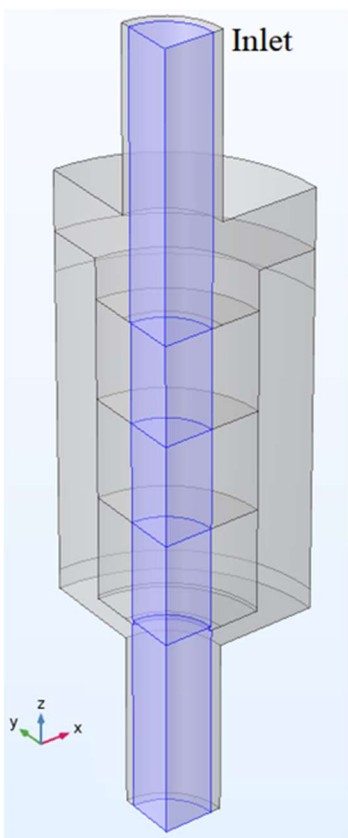

**Figure 3.** The schematic view of the 3D axisymmetric CFD model: a quarter of the experimental setup. Blue domains are inlet, stack of 3 filters and outlet respectively.

The models were constructed according to the actual filters and experimental apparatus dimensions, as shown in Figure 1 and Table 1. In addition, the fluid and porous matrix properties, e.g., fluid density, fluid temperature, dynamic viscosity, Darcy and non-Darcy coefficients and filter open pore porosities were set according to the experimental and empirical data presented in Table 1 and in results chapter as well as explained elsewhere [29]. Then, the CFD simulations were numerically solved using COMSOL Multiphysics® 5.5 software.

### 2.2.1. Assumptions

The following assumptions were considered in setting up the CFD model:

1.  The solution is independent of time, i.e., a stationary solution with initialization was used;
2.  The solution is identical in all quarters of the model, i.e., the simulated experimental apparatus;
3.  The filters have perfectly cylindrical shapes;
4.  The fluid temperature, density and dynamic viscosity are constant;
5.  The pipe surface is smooth, which is expressed by using a no-slip boundary condition;
6.  Gravitational force is not considered (note that, in the experiment, the filters were positioned horizontally).

### 2.2.2. Transport Equations

To choose an appropriate fluid flow module for numerical modelling in the software, the Reynolds numbers were calculated. The Reynolds number ($R_e$) in a pipe can be calculated as follows [39,58,59]:

$$R_e = \frac{\rho V D}{\mu} \tag{6}$$

where $\rho$ (kg/m$^3$) is the fluid density, $V$ (m/s) is the fluid velocity, $D$ (m) is the pipe diameter and $\mu$ (Pa·s) is the fluid dynamic viscosity.

The calculated Reynolds numbers for the well-sealed stacks of three 30, three 50 and three 80 PPI CFFs were in the range of 2400–15,100. The turbulent flow regime begins at $R_e$ > 2300 [39,58]. Therefore, a turbulent fluid flow module was used to mathematically calculate the fluid flow. To be specific, the "Turbulent flow, Algebraic yPlus" module with an added porous media domain was used to simulate the fluid flow in both the fluid and porous domains. It is necessary to mention that the other well-known and commonly used turbulent flow modules, e.g., *k-ε*, *k-ω*, etc., are only available in the normal fluid domains and are not yet included within a porous media domain as defined in COMSOL Multiphysics® 5.5 [60]. As a result, and based on the assumptions made, the following governing transport equations for fluid flow in the pipe sections, as well as the filter media, were solved:

i.   The Reynolds-averaged Navier–Stokes (RANS) equations for incompressible fluids, including the continuity and conservation of momentum equations;
ii.  The Brinkman–Forchheimer equation, together with the continuity equation, for calculating the flow in the porous domains;
iii. An algebraic equation to model turbulence.

More specifically, the governing transport equations can be expressed as follows:

### 2.2.3. The Reynolds-Averaged Navier–Stokes (RANS) Equations

A turbulent flow consists of arbitrary fluctuations of various fluid properties, e.g., velocity, pressure, etc., in time and space [39,61]. It is believed that existing mathematical knowledge cannot handle such random fluctuations [39]. Hence, Reynolds' statistical approach to mathematically simplify the turbulent flow studies is commonly used [39,58–63]. The details of the averaging procedure are provided in the following references [39,61,63].

At steady state and for an incompressible isothermal Newtonian flow, i.e., where the fluid viscosity and density are constant and the stress/strain rate is linear, the Reynolds-averaged Navier–Stokes (RANS) equations for continuity and momentum, in their general form, can be written as follows [39,61,64]:

$$\frac{\partial U_i}{\partial x_i} = 0 \tag{7}$$

$$\rho U_j \frac{\partial U_i}{\partial x_j} = -\frac{\partial P}{\partial x_i} + \rho \frac{\partial}{\partial x_j}\left[\frac{\mu}{\rho}\left(\frac{\partial U_i}{\partial x_j} + \frac{\partial U_j}{\partial x_i}\right) - \overline{u_i u_j}\right] \tag{8}$$

where $\rho$ is density, $U_i$ is the time-averaged mean velocity in $x_i$ direction, $U_j$ is the time-averaged mean velocity in $x_j$ direction, $P$ is pressure, $\mu$ is the dynamic viscosity and $\left(-\overline{\rho u_i u_j}\right)$ is known as the Reynolds stress tensor ($\tau_{ij}$). The term on the left-hand side of Equation (8) represents convection. The right-hand side of the equation includes pressure gradient, viscous diffusion and turbulent diffusion terms.

To close Equation (8), the Reynolds stress tensor ($\tau_{ij}$), which is an unknown term, needs to be modelled [39,63–65]. The Reynolds stress tensor ($\tau_{ij}$) can be expressed as a function of turbulent or eddy viscosity ($\mu_T$) [39,58,60,61,66]:

$$\tau_{ij} = -\overline{\rho u_i u_j} \approx \mu_T \frac{dU}{dy} \tag{9}$$

L. Prandtl [67] introduced a correlation between eddy or turbulent viscosity and mixing length, i.e., Equation (10). In this theory, it is assumed that a lump of fluid which is displaced in transverse direction maintains its mean properties for a characteristic length of $l_{mix}$ before losing momentum and mixing with its surroundings [39,60,61,63,66–68].

$$\mu_T \approx \rho l_{mix}^2 \left|\frac{dU}{dy}\right| \tag{10}$$

Later, Kármán [66] assumed that the mixing length ($l_{mix}$) is proportional to the distance $y$ from the wall, and the proportionality constant $\kappa$, i.e., the Kármán constant, has a typical value of ~0.41 [39,61,66,68].

$$l_{mix} \approx \kappa y \tag{11}$$

### 2.2.4. Brinkman–Forchheimer Equation

The Reynolds-averaged Brinkman–Forchheimer equation for a porous region in steady state and in its general form can be expressed as [43,62,64,69–71]:

$$\frac{\rho}{\varepsilon} U_j \frac{\partial U_i}{\partial x_j} = -\frac{1}{\varepsilon}\frac{\partial P}{\partial x_i} + \frac{\rho}{\varepsilon}\frac{\partial}{\partial x_j}\left[J\frac{\mu}{\rho}\left(\frac{\partial U_i}{\partial x_j} + \frac{\partial U_j}{\partial x_i}\right) - \overline{u_i u_j}\right] - \frac{\mu}{k}U_i - \frac{\rho \varepsilon C_F}{\sqrt{k}}\left[\sqrt{U_j U_j}U_i + \frac{U_j}{\sqrt{U_j U_j}}\overline{u_i u_j}\right] \tag{12}$$

where $\varepsilon$ (percentage) is the filter porosity, $J$ (dimensionless) is the viscosity ratio, $k$ (m$^2$) is the Darcy permeability coefficient and $C_F$ is a dimensionless form drag [36] also referred to as the Forchheimer parameter, Forchheimer coefficient or geometric function [36,38,43,60,64,71–73].

The term on the left-hand side of Equation (12) is convective inertia, and the right-hand side consists of pressure gradient, viscous diffusion and turbulent diffusion terms, as well as the Darcy term and the non-Darcy or Forchheimer term [64]. The Darcy and non-Darcy terms represent the resistance to fluid flow in the porous media [34,73–75].

In the literature [70–72,76], the Forchheimer parameter, or geometric function $C_F$, and Darcy permeability coefficient $k$ can be estimated from porosity and particle diameter using Equations (13) and (14). Here, the open pore porosity is the relevant parameter for fluid dynamics [3] in CFD simulations. However, as defined in COMSOL Multiphysics® 5.5 [60], the permeability coefficient $k$ is specified by the user and is not computed based on porosity and particle diameter as it is for Equation (14). Furthermore, the dimensionless Forchheimer parameter $C_F$ can be defined by the user; otherwise, the default value of 0.55 [60] is used. In 1964, Ward suggested that $C_F$ might be a universal constant with an approximate value of 0.55 [36,38,77,78]. Later, it was found that the dimensionless Forchheimer parameter $C_F$ varies with the nature of the porous medium [36,38,77].

$$C_F = \frac{1.75}{\sqrt{150\varepsilon^3}} \tag{13}$$

$$k = \frac{\varepsilon^3 d_p^2}{150(1-\varepsilon)^2} \tag{14}$$

Recently, it has been shown that the non-Darcy drag term, i.e., the Forchheimer drag term $\beta$ defined in Equation (15), cannot accurately model the experimentally obtained pressure gradient data [26]. More specifically, it was found that the average error could be as high as 87 percent [26]. It was also shown that the average error was as low as 4.4 percent when the empirically obtained Darcy and non-Darcy coefficients of the Forchheimer equation, i.e., the $k_1$ and $k_2$ terms in Equation (2), are used in mathematical modelling [26]. Therefore, Equation (16) was used to model the second-order drag term instead of Equation (15) [2,26].

$$\beta = \frac{\rho \varepsilon C_F}{\sqrt{k}} \tag{15}$$

$$\beta_F = \frac{\rho}{k_2} \tag{16}$$

### 2.2.5. Boundary Conditions

The same boundary conditions as were used in CFD modelling of single filters, as explained elsewhere [2,26,29], were also applied for the CFD modelling of the stacks of three well-sealed samples. Table 2 shows the complete list of the boundary conditions. No viscous stress with the maximum applied pressure was used at the inlet. No-slip conditions for the walls and a uniform outflow velocity were applied at the outlet. The outlet velocity values were based on the obtained experimental data.

**Table 2.** Boundary conditions.

| Inlet | Outlet | Wall |
|:---:|:---:|:---:|
| $p = 50,000 \; Pa$ | $u = U_0 n \; \mathrm{m.s^{-1}}$ | $u = 0 \; \mathrm{m.s^{-1}}$ |

## 3. Results

### 3.1. Liquid Permeability Experiments

The empirically obtained pressure gradients and mean fluid velocity data for the stacks of three commercial alumina ceramic foam filters (CFFs) with 30, 50 and 80 PPI are presented in Figure 4. Each data point in the figure represents an average of a minimum of 35 readings with a confidence interval of 95 percent. As shown in Figure 5 for the stacks of three well-sealed 50 PPI filters, the minimum and maximum margins of errors for the superficial velocities lay in the range of 1.05 to 5.68% and 0.30 to 0.52% for the pressure gradients, respectively.

The first- and second-order permeability coefficients in the Forchheimer equation, i.e., the Darcy ($k_1$) and non-Darcy ($k_2$) coefficients in Equation (2), of the well-sealed stacks of three filters were estimated based on the experimentally obtained mean fluid velocity and pressure gradient data. Ergun's approach, dividing the Forchheimer equation, i.e., Equation (2), by velocity and applying a linear regression [2,19,26,29,79], was used to acquire the permeability coefficients, as explained in Materials and Method (Section 2.1). The empirically obtained Darcy and non-Darcy coefficients, as well as the fluid properties used during the liquid permeability experiments, are summarized in Table 3.

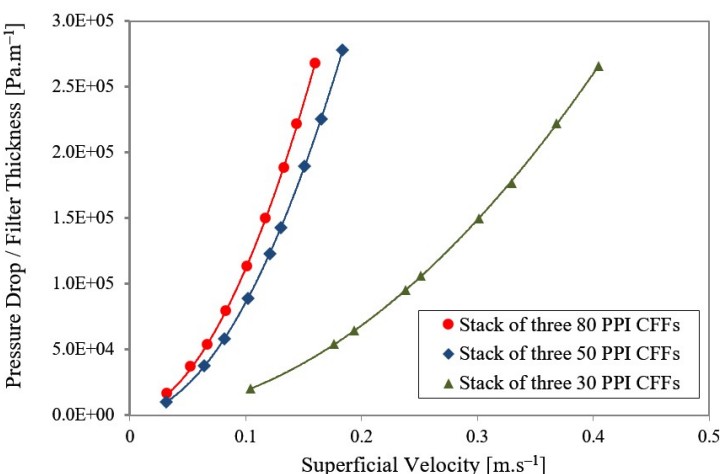

**Figure 4.** Empirically obtained pressure gradients of stack of three 80 PPI CFFs (the red, solid curve), stack of three 50 PPI CFFs (the blue, solid curve) and stack of three 30 PPI CFFs (the green, solid curve) as a function of superficial velocity.

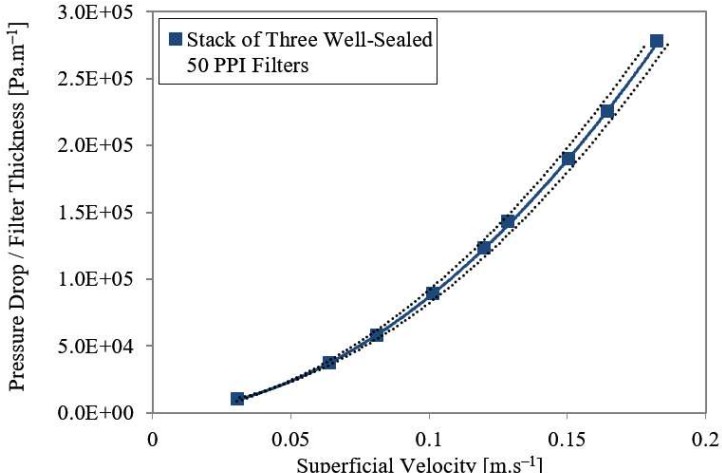

**Figure 5.** Empirically obtained pressure gradients of the stack of three well-sealed 50 PPI as a function of superficial velocity (the dark, solid curves) with a confidence interval of 95% margin of error (the dotted curves).

**Table 3.** The measured and empirically obtained water properties, as well as the empirically obtained permeability coefficients, i.e., the Darcy $k_1$ and non-Darcy $k_2$.

| Sample No. | Water Temperature (K) | Water Viscosity (Pa.s) | Water Density (Kg.m$^{-3}$) | $k_1$ (m$^2$) | $k_2$ (m) |
|---|---|---|---|---|---|
| N1N2N3 30 | 283.7 | $1.28 \times 10^{-3}$ | 999.7 | $3.67 \times 10^{-8}$ | $6.51 \times 10^{-4}$ |
| N1N2N3 50 | 283.6 | $1.29 \times 10^{-3}$ | 999.7 | $1.70 \times 10^{-8}$ | $1.28 \times 10^{-4}$ |
| N1N2N3 80 | 284.2 | $1.27 \times 10^{-3}$ | 999.6 | $6.42 \times 10^{-9}$ | $1.08 \times 10^{-4}$ |

### 3.2. Numerical Modelling

To obtain the optimum mesh configuration, several mesh options were used, and the effect of mesh size on the numerically obtained mass flow rates at given outlet velocities was compared. Table 4 presents a summary of the mesh optimization parameters, including the minimum and maximum element sizes in the domains and boundaries, total mesh elements and calculation times. Figure 6 illustrates the estimated mass flow rate for each mesh option. The computational time and CFD-estimated mass flow rate were used as criteria to select the suitable mesh option. As a result, mesh option 10 was selected to proceed with the CFD modelling of the stacks of three filters.

**Table 4.** Mesh optimization parameters.

| Mesh No. | Element Size in Domains (mm) | | Element Size in Boundaries (mm) | | Total Mesh Element (millions) | Calculation Time (minutes) |
|---|---|---|---|---|---|---|
| | Min. | Max. | Min. | Max. | | |
| 1 | 3.2 | 10.4 | 1.6 | 5.36 | 0.13 | 3.2 |
| 2 | 2.4 | 8 | 0.8 | 4.24 | 0.24 | 5.3 |
| 3 | 1.6 | 5.36 | 0.32 | 2.96 | 0.79 | 18.5 |
| 4 | 0.8 | 4.24 | 0.12 | 1.84 | 2.3 | 63 |
| 5 | 0.32 | 2.96 | 0.12 | 1.84 | 2.9 | 63 |
| 6 | 0.12 | 1.84 | 0.12 | 1.84 | 4 | 95 |
| 7 | 0.1 | 1.5 | 0.1 | 1.5 | 1.3 | 29 |
| 8 | 0.08 | 1 | 0.08 | 1 | 3.9 | 103 |
| 9 | 0.06 | 0.8 | 0.06 | 0.8 | 7.2 | 210 |
| 10 | 0.04 | 0.6 | 0.04 | 0.6 | 16.8 | 580 |
| 11 | 0.04 | 0.4 | 0.04 | 0.4 | 27 | 1273 |
| 12 | 0.32 | 2.96 | 0.016 | 1.04 | 17.5 | 1095 |
| 13 | 0.12 | 1.84 | 0.016 | 1.04 | 21.4 | 1435 |

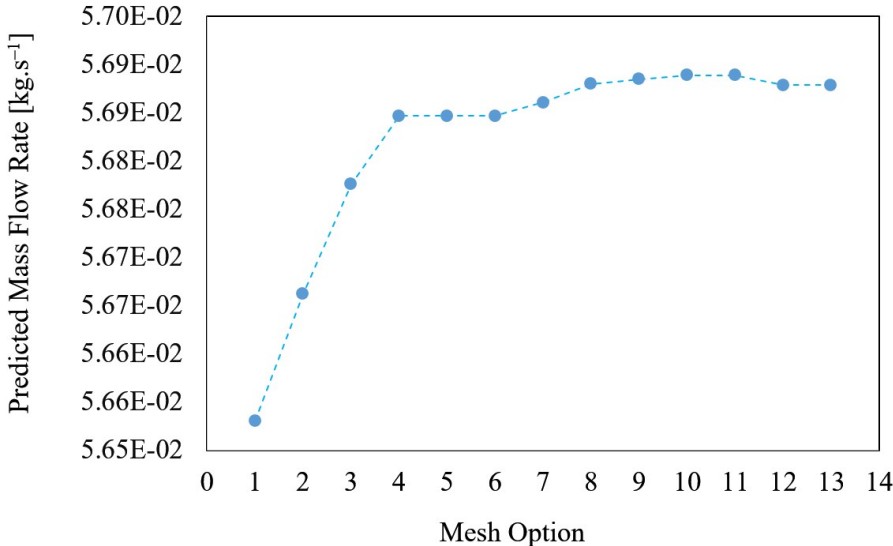

**Figure 6.** Estimated mass flow rate as a function of mesh option.

In total, three models were created according to the filters and experimental setup dimensions as explained in the Materials and Method (Section 2.2). To be specific, one model for each PPI filter was created. As explained in the Materials and Method (Section 2.2) and shown in Figure 7 for a well-sealed 80 PPI model, as well as for other PPI CFFs in this study, the fluid enters the modelled pipe section from one side, flows through the filters and leaves the filters and pipe from the opposite side. The experimentally obtained and mathematically estimated pressure gradients, i.e., the pressure drop over filter length as a function of superficial velocity for the well-sealed stacks of three 30, 50 and 80 PPI ceramic foam filters, are shown in Figure 8. The figure also shows the deviations between the experimentally obtained (the solid curves) and CFD-estimated (the dotted curves) pressure gradients.

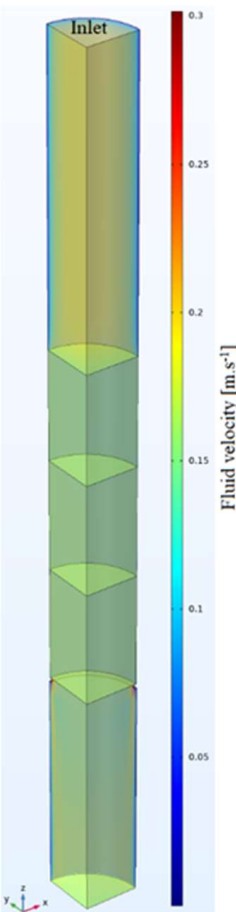

**Figure 7.** 3D view of stack of three 80 PPI modelled CFF at 0.16 m.s$^{-1}$ outflow velocity.

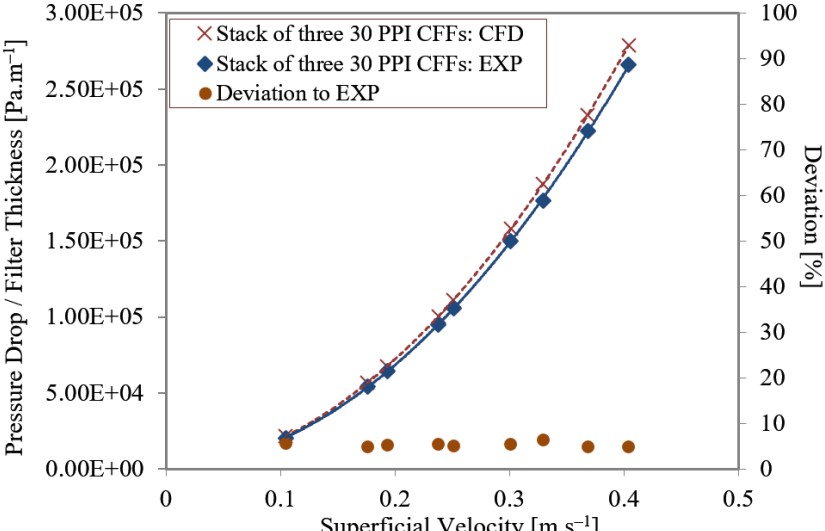

**Figure 8.** *Cont.*

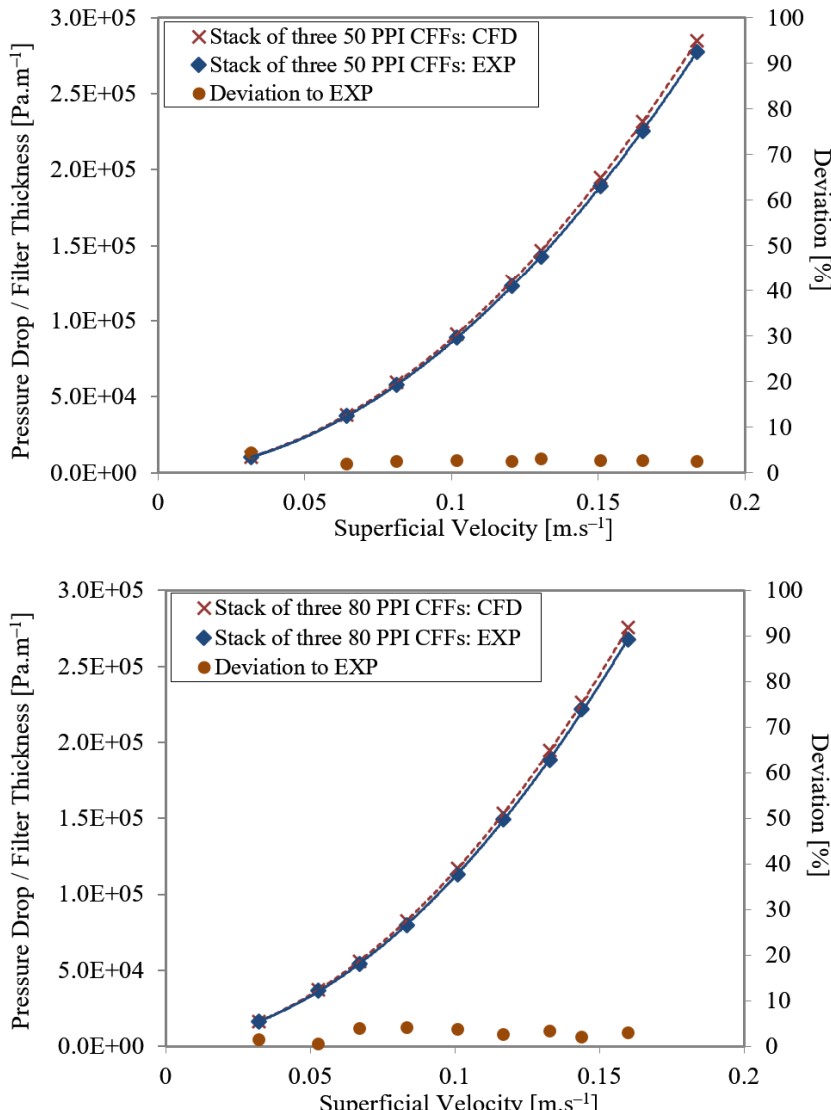

**Figure 8.** Comparison between the experimentally obtained (the solid curves) and the CFD estimated (the dotted curves) pressure gradients of the well-sealed stacks of three 30, 50 and 80 PPI CFFs.

## 4. Discussion

The pressure gradient profiles as a function of superficial velocity for the stacks of three well-sealed [2,20,26,29] 30, 50 and 80 PPI alumina CCFs were experimentally obtained, as illustrated in Figure 4. The higher-grade PPI CFFs contained larger quantities of small cells, windows and strut diameters [19,26,29], as well as lower open pore porosities (see Table 1). These physical properties intensified both the viscous and inertial resistance in the higher-grade PPI filters. Consequently, as shown in Figure 4, an increase in the PPI grade at any given fluid velocity requires a larger pressure gradient.

The recently obtained data were also compared to the findings from previous research works [2,19,26,29], focusing on well-sealed single 30, 50 and 80 PPI alumina CFFs, as shown in Figure 9. As explained in the Materials and Method (Section 2.1), not sealing can cause up to a 60% deviation in the experimentally obtained pressure gradients compared to fully sealed single filters [2]. In the figure, the red, solid curves represent the pressure gradient profiles of the stacks of three filters. The recent experimental procedure is similar to the research performed by Akbarnejad et al. [2,26,29] and Kennedy et al. [19]. The only difference to Akbarnejad's work was the stacking of three 30, three 50 and three 80 PPI CFFs. In recent work, as well as in previous work, the filters were taken from the same manufacturer, were the same size and were from the same location but from different

batches [2,26,29]. On the other hand, Kennedy's work was based on alumina CFF samples from the same manufacturer but from different batches of different-sized commercial filters from different locations [19,26].

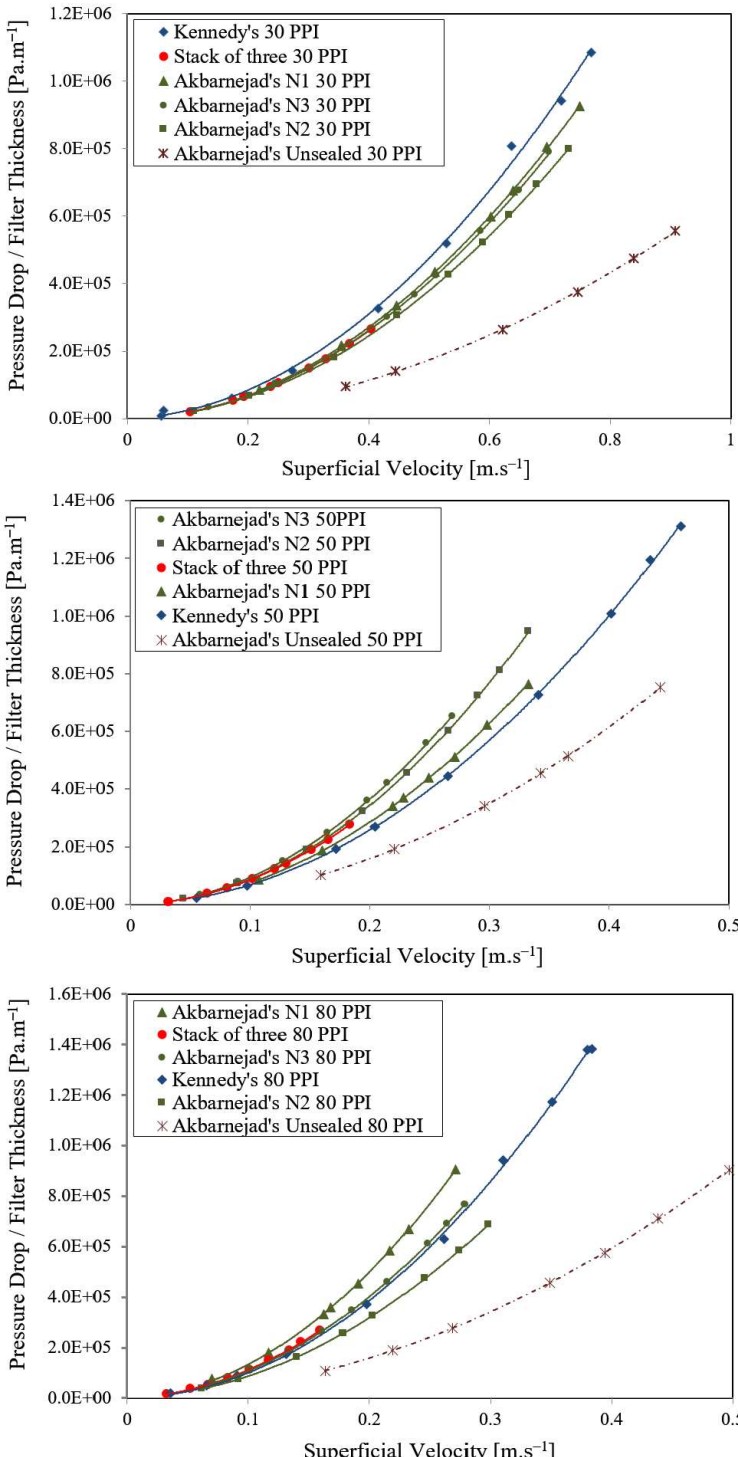

**Figure 9.** Experimentally obtained pressure gradients of stacks of three filters (the red, solid curves) vs. the experimentally obtained pressure gradients of single filters from previous studies by Akbarnejad et al., 2016 & 2017 [2,26,29] (the green, solid curves, as well as the pressure gradients of the unsealed single filters, the dotted curves) and Kennedy et al., 2013 [19] (the blue, solid curve) as a function of superficial velocity.

The Darcy $k_1$ and non-Darcy $k_2$ permeability coefficients of the stacks of three well-sealed 30, 50 and 80 PPI alumina CCFs were also empirically obtained, as explained in Section 2.1 and presented in Table 3. The recent data were compared to the previously obtained data by Akbarnejad et al. [2,26,29], Kennedy et al. [19] and Zhang [57] for single 30, 50 and 80 PPI alumina CFFs, as illustrated in Figures 10 and 11.

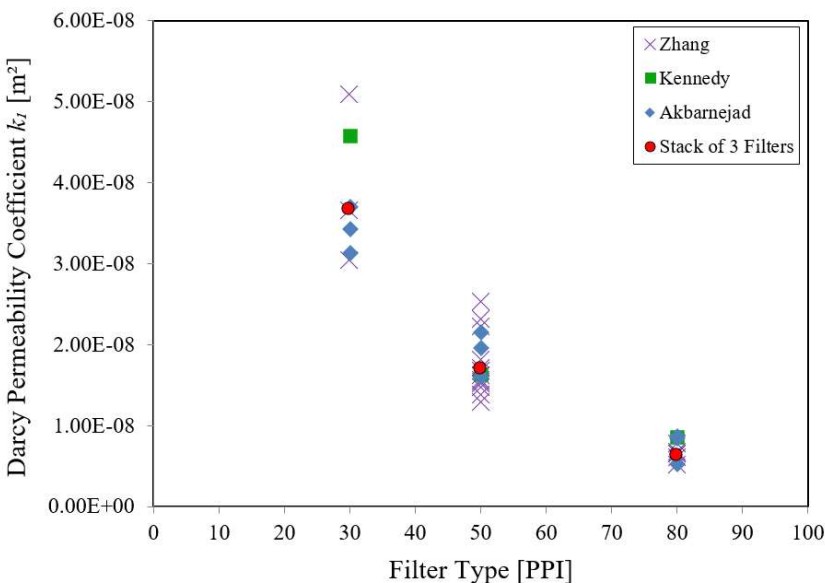

**Figure 10.** A comparison between the empirically obtained Darcy permeability coefficients ($k_1$) of the recent and previous works data from [19,26,57] as a function of filter type (PPI).

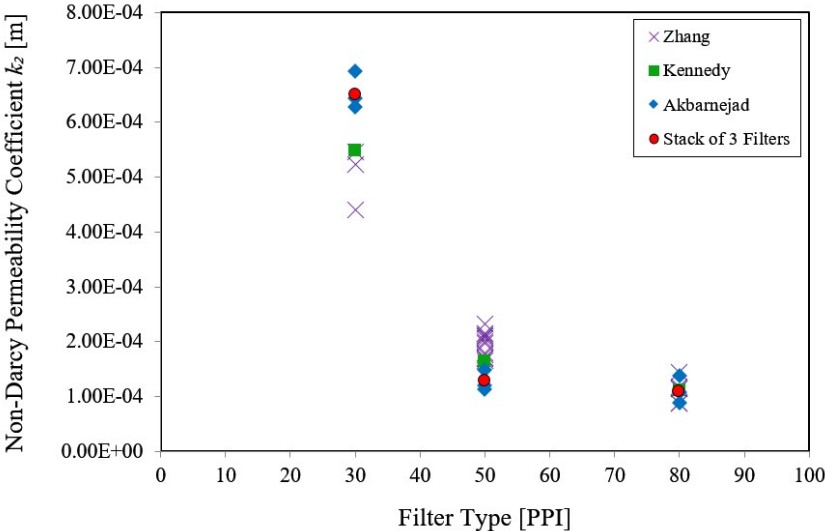

**Figure 11.** A comparison between the empirically obtained non-Darcy permeability coefficients ($k_2$) of the recent and previous works date from [19,26,57] as a function of filter type (PPI).

The pressure gradient profiles of the stacks of three identical alumina CFFs were the same as or near to the pressure gradient profiles of the single well-sealed alumina CFFs, as illustrated in Figure 9. This indicates that, at any given velocity, a nearly identical pressure gradient, i.e., pressure drop over the filter length, can be expected. This is also in agreement with both Equations (1) and (2) in that there is a linear relationship between pressure drop and filter length. It also proves that there was no fluid bypassing, and the sealing procedure was successful. As a result, one may conclude that a three times higher pressure and/or pressure drop needs to be applied in a stack of three identical filters to achieve the same fluid velocity when compared to a single filter of the same PPI. Therefore, approximately

equivalent Darcy ($k_1$) and non-Darcy ($k_2$) permeability coefficients for both stacks of three identical and single alumina CFFs can be anticipated, as shown in Figures 10 and 11. The nature of the deviations are believed to be due to: (i) inevitable variations in the filters' physical properties, including dimensions and open pore porosities, (ii) batch-to-batch variations and (iii) taking samples from different locations of the alumina CFFs.

The calculated Reynolds numbers for the three stacked 30, 50 and 80 PPI CFFs were in the range of 2400–15,100. In such Reynolds number and velocity ranges, the pressure drop becomes nonlinear, and Darcy's law, i.e., Equation (1), cannot describe the flow [13,31,33,34,36–41]. Consequently, for an incompressible fluid flow through a homogenous porous medium, the Forchheimer equation, i.e., Equation (2), can be used [2,6,13,19,26,29,31,36,39,41]. Therefore, a turbulent fluid flow module with an added porous media domain was used to numerically calculate the fluid flow, i.e., Equations (7)–(12). Here, it was possible to either include the Forchheimer parameter or geometric function $C_F$ and Darcy permeability coefficient $k$ estimated based on porosity and particle diameter using Equations (13) and (14) or include user-defined values. However, and as shown recently, the non-Darcy drag term, i.e., the Forchheimer drag term $\beta$ defined in Equation (15), cannot accurately model the experimentally obtained pressure gradient data [26]. More specifically, it was found that the average error could be as high as 87 percent [26]. It was also revealed that when the empirically obtained Darcy and non-Darcy coefficients of the Forchheimer equation, i.e., the $k_1$ and $k_2$ terms in Equation (2), are used in mathematical modelling the average error is as low as 4.4 percent [26]. Therefore, for the numerical modelling of this research work, the empirically obtained Darcy and non-Darcy coefficients, as well as the measured and obtained fluid properties presented in Table 3, were applied for CFD modelling.

The numerically obtained, i.e., CFD-estimated, pressure gradients of the well-sealed stacks of filters revealed a good agreement with the experimentally obtained data, as shown in Figure 8. The deviations of the experimentally obtained data were calculated to be in the range of only 0.4 to 6.3% for all three PPI types of the filter. The bias between the numerically and experimentally estimated pressure gradients is believed to be due to the assumption made in the CFD modelling, i.e., the modelled filters were assumed to have a perfectly cylindrical shape [26,29].

Figure 12 presents a comparison between the mathematically and empirically obtained pressure gradient values for the stacks of three 30, three 50 and three 80 PPI alumina CFFs against the 1:1 diagonal line. The observed level of agreement, only 0.4 to 6.3% for all three PPI types of the filter, between the mathematically and empirically obtained values can also be considered as a confirmation of the adequacy of the CFD modelling.

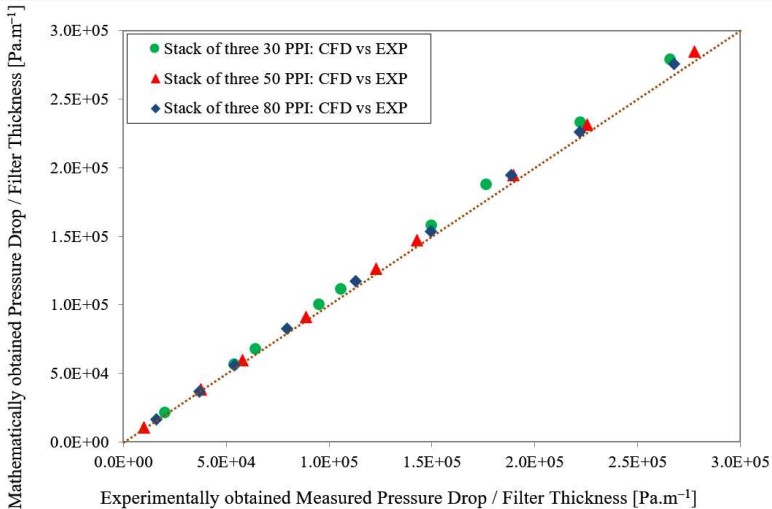

**Figure 12.** A comparison between the mathematically and empirically obtained pressure drops over the filter thickness values for stacks of three 30, three 50 and three 80 PPI CFFs.

## 5. Conclusions

Pressure gradients as functions of fluid velocities for well-sealed stacks of three 30, three 50 and three 80 PPI commercial alumina ceramic foam filters were experimentally obtained. The Darcy ($k_1$) and non-Darcy ($k_2$) permeability coefficients were empirically derived. In addition, three-dimensional axisymmetric mathematical models for each filter grade (PPI) were created based on the experimental conditions, i.e., filter dimensions and porosities and densities and temperatures of the fluid, as well as the fluid flow rates. The numerically obtained pressure gradients were compared to the experimental data. The main conclusions from the recent research can be summarized as follows:

- Stacks of three identical filters from three different batches give substantially the same experimentally obtained pressure gradients as single filters. Therefore, nearly identical Darcy ($k_1$) and non-Darcy ($k_2$) coefficients for a single alumina ceramic foam filter can be empirically obtained;
- As expected, about three times greater pressure and/or pressure drop is required to make the fluid travel through the fully sealed stacked filters of the same PPI compared to an identical single filter;
- The numerically obtained pressure gradients of the three identical filters are in good agreement with the experimental data, and the deviations are in the range of 0.4 to 6.3%.

**Author Contributions:** Conceptualization, S.A. methodology, S.A.; software, S.A.; validation, S.A. formal analysis, S.A.; investigation, S.A.; resources, S.A, A.T. and P.G.J.; data curation, S.A.; writing—original draft preparation, S.A.; writing—review and editing, S.A., A.T., D.-Y.S. and P.G.J.; visualization, S.A.; supervision, A.T., D.-Y.S. and P.G.J.; project administration, S.A.; All authors have read and agreed to the published version of the manuscript.

**Funding:** This research received no external funding.

**Acknowledgments:** The authors wish to express their gratitude to Egil Magne Torsetnes at NTNU for fabrication of the filter holder apparatus. The laboratory and technical support from the materials science departments at KTH and NTNU are also acknowledged.

**Conflicts of Interest:** The authors declare that they have no conflict of interest.

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
