# Peer review of "Effect of Batch Dissimilarity on Permeability of Stacked Ceramic Foam Filters and Incompressible Fluid Flow: Experimental and Numerical Investigation"

_metals, doi:10.3390/met12061001_

Round 1
Reviewer 1 Report
This work investigates the permeability through commercial alumina ceramic foam filters. The numerical results are compared to the experimental data and previous data, and good agreement is achieved. To improve the manuscript, the following concerns can be concerned.
1. The application and significance of this study can be introduced more in Introduction,.
2. The term “batch dissimilarity” in the title is unclear.
3. Some figures can be merged, e.g., the subfigures in Fig. 5.
4. How is the coefficient k2 calculated or measured?
5. The comparison of the Darcy's law with Forchheimer equation can be discussed in terms of the numerical results.
Author Response
Dear Sir/Madam
Thank you for your feedback and time. I have replied to all comments. Please see the attachment.
With regards
Shahin Akbarnejad

Reviewer 2 Report
In my opinion, the contents of the paper do not align with the scope of the Journal "Metals". It addresses the pressure drop in ceramic foam filters - the only connection to metals is that these can be used for filtering metallic melts. It may be better suited for the "sister" journal Materials?
Nonetheless, a few comments that may be considered to further improve the document:
- Table 1 - uncertainty for total and open porosity should be provided.
- Fig. 2 - size markers should be used instead of the rulers which are very difficult to read
- Equation (3) should include units
- It is unclear to me how exactly fluid velocity was measured from the indirectly measured mass flow rate. Specifically, was the reduced cross section in foams and the resulting increase in local velocity considered? All sensors should be clearly indicated in Fig. 1. In general, the description of the experimental procedure needs to be more detailed. How exactly is measurement data processed to eventually obtain the data presented in the Figures of the result section?
- Many of the Figures seem redundant. E.g. Fig. 2 does not add new information to the publication. Similarly, Fig. 3/7 (a) and (b) are redundant and at least one can be removed. Figs. 4 and 5 (and many of the following) can be combined.
- I suggest replacing "mathematical" by "numerical" modelling to emphasize the use of CFD
- The boundary conditions for the outlet should be at least explained at a high level instead of referring to a reference. It is in order to do so for the details but the reader should be able to appreciate the principal idea behind the numerical model.
- I do not understand the rationale behind the numerical analysis. It mainly serves to demonstrate that a commercially available model is able to reproduce experimental measurements with good accuracy? Is there anything else that can be learned?
- Please clarify whether the foams are contacting each other or whether a small gap exists between them.
Author Response

(The authors gave the same response as above.)

Reviewer 3 Report
The paper presents an experiment to obtain permeability coefficients of the ceramic Foam Filters and the result are compared to mathematically modelled data as well as previous research works. Although the manuscript appears valuable it needs some revisions; some suggestions are listed below:
1. The abstract should be more concise.
2. It is recommended to modify Figure 2.
3. In the first paragraph of Results, the author says “As shown in Figure 5, the minimum and maximum margins of errors for the superficial velocities lay in the range of 1.05 to 6.01% and 0.30 to 1.42% for the pressure gradients respectively”. Please explain how you determined the range of error.
4. It is recommended to make explanation of the second conclusion in the manuscript.

Author Response

(The authors gave the same response as above.)

Round 2
Reviewer 2 Report
The authors have addressed all of my comments and the manuscript is now at a level suitable for publication.